# Rationalism or Intuitionism: How Does Internet Use Affect the Perceptions of Social Fairness among Middle-Aged Groups in China?

**DOI:** 10.3390/ijerph19169855

**Published:** 2022-08-10

**Authors:** Yong Xie, Lin Wu, Bo Liu

**Affiliations:** 1School of Sociology, Wuhan University, Wuhan 430072, China; 2Department of Psychology, School of Philosophy and Sociology, Jilin University, Changchun 130012, China

**Keywords:** internet use, social fairness perception, rationalism, intuitionism

## Abstract

Background: In the digital age, the Internet has profoundly affected our production and life, which in turn has affected our mental health. However, little research has been conducted on when and how Internet use (IU) affects social fairness perception (SFP). Methods: Using the data of Chinese General Social Survey (CGSS) 2015, this paper identifies the causal effect of IU on Chinese middle-aged people’s SFP through Ordinary Least Square (OLS) regression and the instrumental variable (IV) method, and uses the Sobel and Bootstrap Test for mediation analysis. Results: IU not only directly reduces Chinese people’s SFP by channeling their social emotions, but also indirectly decreases SFP through the inspiration of government trust. However, inconsistent with some previous studies, social comparison mainly has a partial masking effect on the causality between IU and SFP. Conclusions: The significant negative impact of IU on SFP is the result of the combination of rationalism and intuitionism.

## 1. Introduction

Fairness perception is the subjective evaluation of the fair status quo, which is related to national development and people’s happiness, and is an important prerequisite for political stability and social solidarity. However, in the process of social change, income disparity and inequitable distribution have always existed and even worsened. In 2021, China’s Gini coefficient rose from 0.279 in 1978 to 0.468 [1]. Given the discrepancy between the ideal and reality, the determinants of SFP and their action paths have become the focus of many studies, which has developed numerous theoretical explanatory mechanisms between the continuum of rationalism and intuitionism [2,3].

Rationalism advocates rational human calculation, arguing that people’s judgments of morality such as fairness are usually based on their careful weighing of known information. Representative theories include: (1) Social structure theory. Leventhal [4] pointed out that when people judge fairness, they will consider what kind of fairness principle to adopt in combination with their own motivation, preferences, etc. For instance, the dominant groups with higher social status are inclined to the principle of deservingness in order to maintain the existing distribution pattern, while disadvantaged groups tend to prefer the principle of averaging [5]. (2) Social comparison theory. It maintains that an individual’s evaluation of SFP depends on comparison with reference groups, which is based on the premise that people have access to a variety of information about themselves and their reference subjects in the distribution processes and outcomes [6,7,8]. (3) Fairness Heuristic Theory, that is in the presence of information uncertainty, people obtain inspiration for fairness beliefs through certain other simple clues [9,10]. Subsequently, Van and Lind [11] developed the Uncertainty Management Theory based on this, and gradually transitioned to an intuitionistic explanation, emphasizing the dominant role of individuals’ emotional cues on their SFP when distributional information is unclear [12]. Other representative studies of intuitionism include the emotion-as-information theory proposed by Schwarz and Clore [13] and the emotions of others under ambiguous information advocated by De Crème et al. [14]. All in all, intuitionism asserts that people are not completely rational, and they sometimes judge social justice based on their own emotions or the impulsive reactions of others’ emotions.

With the development of information technology, the digital age centered on the Internet has profoundly changed the face of society, penetrating into people’s production and lifestyle in all directions and influencing their ways of thinking and values. The Internet has provided us with a broad platform for online communication and entertainment, helped to increase our social support and social capital, reduced loneliness and depression, and improved subjective well-being and life satisfaction [15,16,17]. On the other side of coin, its improper use, or “Internet addiction” or “technology dependence” may lead to sleep anxiety and aggravate depression levels, which is detrimental to people’s mental health [18,19,20]. However, little attention has been paid to the impact of the Internet on SFP. Indeed, theoretically, we receive a great deal of information on the Internet every day and are infected by others’ emotions, all of which may have some impact on an individual’s SFP.

One line of research closely related to our work looks at the role of new media in influencing the SFP. Yao [21] pointed out that neither the type of media use nor the frequency of Internet access affected college students’ SFP. However, the empirical results of Lu et al. [22] and Han and Xie [23] showed that the use of new media had a catalytic effect on the decrease of SFP. In addition, a few scholars have noted the role of the Internet, a specific medium type, on the SFP. Lu and Duan [24] used the data of CGSS 2010 and found that the higher the frequency of Internet use, the lower the SFP among youth. Furthermore, based on the comparison between traditional media and new Internet media, Zhu Bin et al. [25] found that on the one hand, the Internet media strengthened people’s perceived social injustice by weakening the “gatekeeper” effect; on the other hand, it enhanced people’s sense of social fairness and personal fairness by providing more opportunities for downward comparison. Zhu et al. [26] used the Endogenous Ordered Probit model to deal with potential endogeneity and advocated that the Internet usage would lead to upward comparisons among farmer groups, which in turn resulted in a lower SFP.

Existing scholarship has not yet reached a consensus on the masking or mediating effect of social comparison, and mainly explains the causal effect of the IU on SFP from a rationalist perspective, lacking an intuitionist perspective and severing intuitionism from rationalism. In addition, the middle-aged group have large differences in lifestyle and psychological characteristics compared with the youth and the elderly, but few researchers have paid attention to this and treated them as an independent research object. In fact, the middle-aged have gradually settled down after the vicissitudes of life, and have a deeper understanding of the SFP. Therefore, in terms of the research objects, this paper focuses on the relationship between the frequency of IU and SFP among middle-aged people; in terms of the research perspective, it insists that SFP is the result of both rationalism and intuitionism, considers the mechanism of the influence of IU on SFP by combining rational and emotional factors, and clarifies the role played by social comparison in it.

## 2. Theories and Hypotheses

### 2.1. Intuitionism—IU, Social Mood and SFP

Intuitionism emphasizes that emotional factors such as one’s own or others’ emotions and intuitions play an important role in the judgment of SFP in a situation of information uncertainty [27,28], with positive emotions producing higher fairness ratings and negative emotions potentially producing lower evaluations [13]. Undoubtedly, the media, as our main source of information, not only provides us with information, but also guides our social attention and social emotions, and influences our SFP. The media deletes, adapts, and processes an existing social phenomenon to create a “mimetic environment” for the public, in which the public in turn forms different cognitions in relation to their own social-cultural background. This is what Lippmann proposed is the way the media influences public’s social cognition—the media transforms “social reality” into “symbolic reality”, and leads to people’s “perceived reality” only reflecting the “symbolic reality”, rather than being a true reflection of “objective reality” [29]. On this basis, Gerbner et al. [30] developed cultivation theory, showing that mass media had huge impact on people’s values and mindsets, and it also worked in China [31,32]. Additionally, even human’s sensitivity to social issues depends to a certain extent on the amount of media coverage [33].

The immediacy, rapidity and convenience of Internet far exceed those of traditional media, greatly enriching the sources and types of information available to people who increasingly rely on the Internet to gather information. However, compared with traditional media, the “gatekeeper” effect of the Internet is also more difficult to maintain. Generally speaking, media such as television and newspapers are subordinate institutions of the established structure, main function of which is to maintain and stabilize the existing social order [30], while the Internet has disintegrated the gatekeeper’s power, leading to the exposure of conflicts and contradictions between the old and the new system, and a spurt of negative information such as the failure of rules and security crises. In such an information-overloaded network society, our sensitivity to information is declining, and “information expectation” has gradually evolved into “information curiosity” [34]. In order to gain attention and seize traffic opportunities, related media often exaggerate and polarize the events to guide the public’s social care. People’s social cognition is gradually distorted, which progressively constructs online social emotions after being rendered by media and individual empathy.

Therefore, we propose hypothesis 1 that the more frequently the Internet is used, the lower the sense of social fairness of the middle-aged groups.

### 2.2. Social Comparison Theory—IU, Social Comparison and SFP 

Social comparison theory assumes that we have a clear understanding of information and develop either a higher or lower sense of relative deprivation and social fairness in our own comparison with a reference group, emphasizing the inequality that people recognize [35]. Based on the similarity and accessibility of reference groups, individuals pose three types of comparisons—horizontal comparison, i.e., comparing with peers or social groups within a certain area [36]; vertical comparison, i.e., comparing with someone’s own or his family’s past experiences [37]; and anticipatory comparison, i.e., comparing with someone’s own expectation [7]. The Internet mainly broadens the scope of our horizontal comparison, and we can even obtain dynamic information about many strangers from the Internet platform, so that the social comparison in this article mainly refers to the horizontal comparison. This individual-based strategy choice may have two consequences: for one thing, the Internet triggers more upward comparisons, and the variety of “ostentatious wealth” on the Internet reduces our subjective status perception and increases our sense of relative deprivation; for another, individuals are motivated by self-protection to prefer downward comparisons, and lower their reference points, thus obtaining higher self-esteem and a more positive evaluation of social equality [38]. We believe that the middle-aged who face heavy life and economic pressure would be more likely to adopt defensive strategies to avoid comparison when using the Internet for relaxation and entertainment, or to compare themselves with people who are worse off than they are, in order to gain temporary gratification and pleasure.

Therefore, we propose hypothesis 2 that the higher the frequency of IU, the more the middle-aged groups tend to compare downwards and develop a higher SFP. In other words, social comparison exerts a masking effect between IU and SFP, buffering the negative impact of IU on SFP.

### 2.3. Fairness Heuristic Theory—IU, Government Trust and SFP

Lind [9] and Van [10] supposed that individuals have a fundamental social dilemma in organizations. If they choose to cooperate with the organization, they may encounter exploitation and oppression by authority; and if they give up the cooperation, they also give up their identity in the organization and its benefits. Then, they need to find organizations they can trust, but because of their lack of comprehensive understanding of the organization, they can only use cognitive shortcuts to effectively deal with uncertainties in the environment. Hence, their judgments of fairness are influenced by the interpretability of information as well as the order of information [39,40] and generate cognitive resistance after forming an impression of fairness until some unexpected event is prompted that might make them revise their concept of fairness.

Actually, the predictive effect of trust on procedural and distributive justice has long been discovered [41,42], because in an asymmetrical information environment, we can only rely on certain trusted institutions to obtain information and form our cognition of events accordingly [43]. The more we have confidence in the government, the more likely it is to use official media as the primary source of information, the public is much less likely to be exposed to negative information and negative emotions, and therefore may have a higher SFP. However, Shah et al. [44] argued that the Internet’s entertainment nature caused a crowding-out effect on political participation and interest, which may bring about the negative effect of “depoliticization”. Under the influence of the Internet, our political knowledge storage and political participation ability are degraded, and the information on the Internet, whether true or false, steadily distorts our impressions of the government, ultimately leading to a decline in our reliance on the government [45,46].

Based on the above discussion, we propose hypothesis 3 that IU reduces an individual’s SFP through lower down the government trust. In other words, government trust mediates the relationship between the middle-aged groups’ IU and SFP, and is a potential mechanism by which IU negatively affects SFP.

## 3. Data and Methods

### 3.1. Data Source

The data used in this article come from the China General Social Survey in 2015. CGSS started from 2003 and is the earliest national, comprehensive and continuous academic survey project in China, which systematically collects data at multiple levels of society, community, household and individual. CGSS 2015 adopted multi-level probability proportional to size (PPS) random sampling. At the village and residential level, based on map addresses, 478 communities nationwide were mapped and sampled. The final survey covered 478 villages and residences in 28 provinces/cities/autonomous regions of China, and collected 10,968 valid questionnaires. Because of focusing on the middle-aged group, we only select the sample data from the age group of 35–59. After deleting and processing some missing and singular values, the remaining effective sample size is 4238.

### 3.2. Variables

The dependent variable is SFP. Although there are multiple dimensions of SFP, in people’s real lives, considering fairness only from a certain decision event cannot fully explain people’s behavior [47], and what really drives people’s behavior is the overall SFP [48]. Hence, we here mainly adopt the overall SFP. Corresponding to the question “In general, do you think the social fairness today is unfair”, the answer items are scored on a five-point Likert scale, from 1 “completely unfair” to 5 “completely fair”. The higher the score, the higher the individual’s SFP. Our independent variable is IU, which corresponds to question “In the past year, how often did you use the Internet (including mobile Internet access) is…”. Similar to SFP, the responses are scored on a five-point scale ranging from 1 “never” to 5 “very often”, and the higher points indicate more frequent use of the Internet.

In order to examine the role of social comparison in the relationship between IU and SFP, we use the personal subjective status (PSS) “Compared to your peers, what do you think your socioeconomic status is…” and the family subjective status (FSS) “What is your family’s economic status in which bracket in the location?” to construct the concept of horizontal comparison. Since PSS was selected from a small sample of 1 “higher”, we recode 1 “higher” and 2 “similar” as 0 to indicate a lower sense of relative deprivation and recode 3 “lower” as 1 to manifest a higher sense of deprivation. As for FSS, due to the small sample size of 5 “much higher than the average”, we combine 4 “above average” and 5 “much above average” and reverse-code answers so that the larger the value, the lower the FSS and the higher the relative deprivation.

To explore the underlying mechanism of the relationship between IU and SFP, we use government trust as a potential mediating variable. In the Chinese context, when discussing trust structure, scholars mostly cite the trust research of Mayer et al. [49], which is precisely from the basic interpersonal trust to understand the trust structure and the basic logic of the dynamic change of trust and can be used for Chinese culture of trust [50]. According to Mayer et al. [49], ability, benevolence and integrity are three key dimensions that affect trust, and they are also important measures of government trust. Among them, ability, reflecting the government’s work performance, requires the government to make a difference in its own area of governance expertise, so it can be measured by the question “Are you satisfied with the government’s performance in the following aspects of work?”. Benevolence requires the government to transcend its self-interest motives and put the needs of the people ahead of personal goals. That is to say, government workers serve the people wholeheartedly, have no special interests of their own, and work with honesty and self-discipline. It is measured by the question “How do you rate the probity of the following types of party and government officials?”. Integrity has two meanings, one of which is consistency, i.e., the formulation of policies is consistent with the specific implementation. It corresponds to “Do the government departments concerned strictly follow the law in their enforcement practices in real-life situations?” The second is impartiality, which requires the government to treat every citizen equally and without bias and is measured by the two questions “Considering all aspects, how satisfied are you with the current public services in China in general in all aspects (the degree of balanced distribution of public service resources)/ (the degree of inclusiveness of public services)? Very dissatisfied, not very satisfied, not satisfied or dissatisfied either, relatively satisfied, or very satisfied?“ The Cronbach alpha value is 0.794, which indicates sufficient reliability and consistency. In order to make the data concise and easy to calculate, we, respectively, calculate the mean answers of the above four variables as their own total scores (some answers need to be reverse-coded first).

In addition, based on the experience of previous literature, we select gender (0 = female), age, residence (0 = rural), region (0 = western), married (0 = unmarried), political affiliation (0 = non-communist), education (0 = high school and below), annual personal income, employed (0 = unemployed), vertical comparison (0 = lower), expected comparison, and subjective well-being (SWB) as control variables.

However, even after controlling for these variables, we still cannot definitely rule out the endogeneity of the independent variable. Therefore, in order to identify the exact causal relationship, we select “average frequency of IU in the same community (AFIU)” as the IV. This variable is similar to IU and is scored on a five-point scale with the larger the value, the higher the frequency. Theoretically, the average frequency of IU in the community is closely related to individuals’ frequency of IU, but in the community, as a factor outside the individual system, the average frequency of IU is unlikely to affect the individual’s SFP. Therefore, AFIU satisfies the two premises of IV, i.e., correlation and exogeneity.

Table 1 presents the specific descriptive statistics for the above variables. The SFP among the middle-aged is generally at a moderate level, with a specific mean value of 3.11. Among them, 1317 samples consider unfair (including “completely unfair” and “relatively unfair”), accounting for 31.08% of the total sample; 1959 samples believe it was fair (including “relatively fair” and “completely fair”), accounting for 46.22%; and 962 samples allege it was “not fair but not unfair either”, accounting for 22.70%. That is to say, most people’s SFP is average or above, but at the same time there are 6.80% of the sample claiming that it is “completely unfair”, and these extreme situations must be taken seriously into account. As for IU, its mean value is 2.36, which is in the lower middle of the 5-point score, indicating that the middle-aged are not adequately familiar with the Internet. Precisely, there are 2100 people in the sample who never use the Internet, accounting for nearly half of the total sample (49.55%); meanwhile, another 10.05% of the sample rarely use the Internet. This may be due to their lack of access to the Internet or their limited digital capabilities and fear of accessing the Internet, which ultimately leads to their temporary failure to integrate well into the digital society.

### 3.3. Empirical Strategy

As long as the regression equation is set up correctly, whether the dependent variable is regarded as a continuous variable using OLS regression analysis, or it is regarded as an ordinal variable using the Ordered Probit or Ordered Logit regression, the direction and significance of the estimated coefficients are consistent, and there is no difference between the two methods. Therefore, in this study, we regard SFP as a fixed-distance variable, and use the multivariate OLS regression to estimate the impact of IU on the middle-aged group’s SFP:Y = b_0_ + b_1×1_ + b_2_X_i_ + … + b_i_X_i_ + v(1)
where Y represents SFP, X_i_ denotes each independent variable, b_i_ means the effect size of each independent variable, b_0_ is the intercept, and v signifies the random error. Given the endogeneity of the independent variable, the estimates of OLS regression may be biased, so we need to further apply the IV for statistical analysis, specifically comprising two stages:Estimate in the first-stage:
X_1_ = α_0_ + α_1_Z_1_ + α_2_X_2_ + … + α_i_X_i_ + u(2)
Estimate in the second-stage:
Y = β_0_ + β_1_X_1_ + β_2_X_2_ + … + β_i_X_i_ + μ(3)
where Z_1_ is the IV of X_1_ IU, which in this paper refers to the “AFIU”; X_1_ in the second stage is estimated from the first stage; X_2_-X_i_ are the remaining exogenous control variables, α_i_ and β_i_ denote the influence coefficients corresponding to each variable, and u and μ symbolize random error terms.

## 4. Results

### 4.1. Correlates of SFP among the Middle-Aged in China

Table 2 presents the estimation results of the OLS regression. In the specific analysis, considering the possible heteroscedasticity, we use the “OLS + robust standard error” method. In Model 1, we regress SFP on IU without including other covariates. Then, in Model 2, we added other covariates. In Model 3, we added family and personal subjective status on the basis of Model 2. The three estimates show that IU plays a negative role on SFP. Specifically, the estimates of Model 3 imply that if IU increases by 1, the SFP may decrease by 0.060 under the premise of other conditions remaining unchanged. It preliminarily confirms hypothesis 1, i.e., negative emotions on the Internet reduce the SFP of middle-aged groups.

With respect to other covariates, since age is significantly correlated with SFP in the correlation analysis while the square of age is not, we choose to include age in the statistical equation (if interested, the specific correlation analysis results can be requested from authors). The results show that the effect of gender, political affiliation and work status is not significant. However, inconsistent with previous studies [23], the effect of age is also insignificant. This may be caused by the different age ranges of the sample groups. Compared with the all-aged, the middle-aged are more consistent, their thoughts are more mature and stereotyped, and their social attitudes are generally less likely to change observably with age. In addition, unmarried and middle adults have higher SFP than those who are married. In terms of residence, rural areas have higher SFP than urban areas. Both the eastern and central regions—relatively more developed regions—have lower SFP than the western regions. In other words, the more economically developed regions have lower SFP, this is because in these regions, there is a mismatch between the realistic fierce competition and the ideal good expectations, and it is difficult for individuals to rise in social status. They create a huge psychological gap and their SFP reduces. The statistical analysis of education and annual income is consistent with the “structural theory” [51]. As reflections of an individual’s position in the social structure, increase in either annual income or education may contribute to greater SFP. The effect of SWB also suggests that social attitudes are positively related to SFP. The parametric results at the comparative level point out that individuals with lower sense of deprivation have higher SFP, either in horizontal comparisons, vertical comparisons or expected comparisons, which is consistent with the “relative deprivation theory” [52].

### 4.2. Impacts of IU on SFP and Its Distribution

To tackle potential endogeneity issues, we further use AFIU as IV for IU. Before using the IV method, we regress SFP on the IV to explore whether AFIU exert a direct impact on SFP. As shown in the first and second columns of Table 3, even including IU and other covariates, the AFIU has no significant impact on SFP. At the same time, the results of the variance inflation factor show that there is no multicollinearity between variables (VIF = 1.54).

The 2SLS estimates are then reported in Columns 3 to 6 of Table 3. The first-stage regression results show that IV has a positive impact on IU with a value of 777.70 for the Kleibergen–Paap F statistic, rejecting the null hypothesis of the weak instrument. Therefore, under the double verification of statistical results and the previous theoretical deductions, the AFIU is a very plausible IV. After excluding endogeneity issues, the impact of IU on SFP remains significantly negative. Using the Internet is associated with a decrease in middle-aged adults’ SFP by 0.096 points. This estimated coefficient appears to be a little different from the result of the OLS regression (0.060), but the Hansen J test for instrument exogeneity (*p* = 0.22) indicates that the two results are not significantly different. Therefore, we believe that the difference stems from the fact that the two measure different sample groups, with the IV method measuring the local average treatment effect [53], that is, our IV identifies only the average effects of those who comply with the assignment-to-treatment mechanism implied by the chosen instrument [54].

We explore the impact of IU on the distribution of middle-aged adults’ SFP by the quantile regression method, setting the quantile range as 0.01–0.7 and the interval to 0.01. The results are shown in Figure 1, where the effect of IU is only significant at quantiles 0.23–0.63. This means IU only has a negative impact on middle-aged individuals with a SFP at or below the average level. This effect hardly further reduces the attitudes of the middle-aged who have extreme views on SFP. Meanwhile, individuals with high SFP are not sensitive to the influence of IU either, that is, the higher the evaluation of social fairness, the stronger and more difficult it is to shake their ideas.

### 4.3. Sensitivity Analysis

In this section, we conduct a battery of sensitivity tests from three perspectives. In column 2 of Table 4, we treat SFP as an ordinal variable and perform an IV-Ordered Probit model for statistical inference. The results reveal that the effect of IU is significant. In the column 3, we recode SFP as a dichotomous variable by combining the answer items of “completely unfair” and “relatively unfair” into “unfair” with a value of 0, and combining “completely fair” and “relatively fair” into “fair” with a value of 1. Furthermore, in order to avoid biased values, the samples that choose the response item “not fair but not unfair either” are deleted. Similarly, in terms of IU, we recode “never” as “no”, taking the value of 0, and combine the rest of the answers into “yes”, taking the value of 1. The statistical results of the IV-Probit model are still significant. Again, considering that the age range of the middle-aged has not yet been determined in the academic community, in the columns 4–6, we replace the sample groups of different ages (40–59 years old; 45–59 years old; 35–55 years old) for estimation. Additionally, in Figure 2, we conduct quantile regression analysis on these three models and also obtain very similar results.

### 4.4. Masking Effect

Notably, comparing the parameter estimations in columns 2–3 of Table 2, we find that the inclusion of the horizontal comparison variable instead increases the negative impact coefficient of IU (−0.054 → −0.060) and it indicates that the horizontal comparison plays a masking effect in it. To examine it, we further perform statistical inferences through the Sobel test and Bootstrap test with the bootstrap sampling times being 1000. The results are shown in Table 5. In path A, the more frequently the Internet is used, the lower the relative deprivation of the middle-aged in the horizontal comparison about themselves and their families. In path B, the horizontal comparison is negatively connected with their SFP. If an individual produces a lower sense of relative deprivation in a horizontal comparison, then his SFP may be higher. Therefore, we can draw the following conclusions: in the process of using the Internet, the middle-aged generally tend to compare downwards out of self-interest motives, which reduces their relative deprivation and enhances their SFP to a certain extent. In other words, social comparison masks the negative effects of IU on SFP, which is consistent with the results of Zhu Bin et al. [25]. In summary, hypothesis 2 is confirmed.

For those individuals with lower economic income, although they are motivated by self-protection, there are few objects on the Internet that can be provided for them with downward comparisons. Faced with the great life presented by others on the Internet, they may develop a higher sense of relative deprivation and more negative evaluation of social justice. For the higher-income groups, they have strong self-interest motives and hope to motivate themselves to work hard through upward comparison, so as to maintain and even obtain further upward mobility [55]. Thus, although they have many opportunities for downside comparison, they lack the enthusiasm of downward comparison, and they have negative SFP due to the solidification of social mobility in upward comparison instead. For the middle-income groups, the Internet expands their pool of downward comparison, and at the same time, they have the motivation of downward comparisons, thereby increasing their self-satisfaction and self-esteem [56]. Therefore, the Internet is likely to have a negative effect among the low-income groups and high-income groups. To test the above hypothesis, according to the definition of Li and Xu [57], we clarify individuals with annual personal income of RMB 35,000 or less into the low-income group, individuals with income over RMB 120,000 into the high-income group, and those in between into the middle-income group. The results in Table 6 are in general agreement with the above theory. For low-income and high-income middle-aged individuals, the influence coefficient of the IU on SFP is significantly negative, while for middle-income individuals, the estimated coefficient is positive but insignificant.

### 4.5. Mechanisms

Based on the existing literature, we regard government trust as a potential intermediary mechanism, and select four potential intermediary variables under it. These four variables are the public’s evaluation of the government’s ability, benevolence, consistency and impartiality. According to the descriptive statistics in Table 1, the people’s perception of the government’s ability is relatively the highest, followed by benevolence, impartiality, and consistency. However, in general, the evaluation of those four aspects is not high, belonging to the medium level.

In the statistical analysis of the mediated effects, we apply the Sobel and Bootstrap methods as well, again setting the Bootstrap sampling number to 1000. Table 7 reports the results, which reveal that, conditional on other covariates, IU exerts a significant negative impact on the government’s ability and impartiality, while it has no significant impact on government benevolence and consistency. This means that while the government’s ability and impartiality may serve as potential mechanisms, the negative impact of IU on SFP is largely not a result of decreased benevolence and consistency. Subsequently, in paths B and C`, the government’s ability and impartiality pass the significance test again. Whether it is the *p*-value test in the Sobel or the confidence interval test in the Bootstrap, the indirect effects of the government’s ability and impartial behavior are significant, but both are only partially mediating. The mediating effect of the former is −0.016, accounting for 26.15% of the total effect, and under its mediation the direct effect of IU is −0.046, while the mediating effect of the latter is −0.005, accounting for 9.09% of the total effect, and the direct effect of IU is −0.055. Totally, the indirect effect of both is −0.021, accounting for 35.24% of the total effect. Therefore, the government’s ability and impartiality play a partial mediating role between IU and SFP, reinforcing the negative causal effect of IU on SFP, so that hypothesis 3 is partially verified. The Internet will increase the possibility of the middle-aged being exposed to negative public information, making them dissatisfied with the performance of the government and believe that the level of equalization and inclusiveness of public services provided by the government is not enough. Once they have doubts about the competence and fairness of the government, their trust in the government also declines, and they question official information and social justice. Then, the cultural structure of trust changes, and people rely more on the social media [58]. At the same time, without the supervision of “gatekeepers”, they are more exposed to an unfair environment and become more disappointed with the government, thus forming a vicious circle and ultimately contributing to a decline in the SFP.

However, it is worth noting that the IU has a positive but insignificant impact on the perception of benevolence, in complete contrast to the study by Deng and Liu [59]. We suppose that this is due to the particularity of the middle-aged. They are more introspective and have a more realistic assessment of society and life, and thus have a certain degree of tolerance towards corruption. Regarding the disclosure of corruption cases on the Internet, they are more likely to regard it as an expression of the government’s anti-corruption efforts, rather than impulsively and simply characterizing it as evidence that the government is full of corruption. As a result, they are instead more likely to believe in the cleanliness of the government as Internet use increases.

## 5. Discussion

This study mainly adopts the following four strategies to examine when and how IU affects Chinese people’s SFP. First, we use the IV method to remove the potential endogeneity of the independent variable and to investigate the causal impact of IU on middle-aged people’s SFP. Second, we explore the potential asymmetry in the causal influence of IU on the distribution of SFP by employing quantile regression. Third, the robustness of the impact is tested by changing different estimated models and sample groups. Based on the above three steps, we confirmed the first core question of the study, that the IU plays negative influence on the SFP of the middle-aged. Finally, we explore the roles of social comparison and government trust in causality through adopting the Sobel test and Bootstrap test, answering the second core question, i.e., how IU affects SFP in middle-aged groups.

We find that IU negatively affected the SFP among the middle-aged in general through the mobilization of social attention and social emotions, which is a direct path. The Internet has broken through the boundaries of “gatekeeper”, and a vast amount of information is flooded in the network, leading to information redundancy, resulting in people being in a mixed information environment for a long time. Vague and negative events spread wantonly in the public space, and gradually form negative social sentiments under the guidance of social media, which is not conducive to people’s evaluation of social fairness. Precisely, the impact of IU varies substantially across quantiles of the SFP distribution. The negative impact of IU is only significant and strong in the lower quantiles of the distribution (0.23–0.63). This suggests that the middle-aged with a low SFP are more likely to be negatively affected by IU, and in turn to bring about a lower SFP. Practically, individuals with more in-between attitudes and vague orientations are more disturbed by IU. However, at the same time, the effect of IU saturates when SFP is below the 0.23 quantile or above the 0.63 quantile. That is to say, the impact on middle-aged individuals outside this range is limited.

In addition to the direct influence of subjective emotions, the Internet also exerts an indirect impact on the SFP of the middle-aged through social comparison and government trust, which tend to be rationalistic paths. Social comparison mainly plays a masking effect, while government trust plays a partial mediating role. Although the Internet has broadened the range of horizontal comparison for people, most of them tend to compare themselves downwards for self-interest motives and their sense of relative deprivation is low, thus increasing their SFP. However, this path does not change the negative effect of IU. Moreover, when the sample is further divided into three income groups—low, middle and high, IU has a significant negative effect on the middle-aged with low or high income. The reason may be that, compared with the middle- and high-income groups, the low-income groups lack opportunities to make downward comparisons, and they are mainly in contact with people who seems to live better than them in the online community, and possibly form more upward comparisons. Thus, they create a higher sense of relative deprivation and a lower SFP. As for high-income groups, they usually have higher expectations and are more expected to motivate themselves to achieve upward mobility through upward comparisons. Therefore, they also develop a higher sense of relative deprivation and a lower SFP in the process of comparison.

We have no evidence that IU reduces middle-aged people’s evaluation of the government’s consistency and benevolence, thereby reducing their SFP. However, it is clear that IU causes a decline in middle-aged people’s satisfaction with the government’s ability to work and a distrust of the government’s impartiality in providing public services. The absence of any one of the three components of trust will decrease the trust [49]. The middle-aged have been informed of numerous passive incidents through their use of Internet, which makes them doubt the ability and impartiality of the government, and decreases their trust and reliance on government. In the case of uncertain information, they are less likely to have a higher SFP relying on government trust. Moreover, the decline in trust in the government makes it more difficult to maintain the official “gatekeeper” role and to control information. The public gradually develops a stereotype of social injustice and government inability, and the government trust is reduced again, and the “gatekeeper” effect is further undermined, and thus forming a vicious circle.

This study has certain policy implications. In recent years, the Chinese government has been making every effort to promote the level of equalization of public services between urban and rural areas and between regions, and the quality of public services and social distribution have been further improved. However, the state’s work in image construction and expression still needs to be strengthened. The Internet, which has the nature of two-way communication, has widened public opinion channels for it. The government should seize this opportunity to build a great image of integrity, honesty and frankness in front of the public, vigorously publicize the “good news”, and at the same time not hide the “bad news”. It has proved that covering up the injustice of events may cause the public’s emotional reaction to change from positive to negative [60]. In addition, in order to prevent IU from further eroding the SFP, the government must take action to regulate the Internet and other new media, purge unethical practices on the Internet, and severely crack down on deliberate gimmicks and out-of-context quotations for the sake of speculation, creating a warm and rational voice for news. At the same time, strengthen training for the public in digital literacy and legal literacy, so that they are capable of knowing right from wrong and making more rational judgments.

There are still some shortcomings in the research: Firstly, the CGSS project team has released the data of CGSS 2017, but its questionnaire is quite different from that of CGSS 2015. It adds new information about the Internet, but simplifies many questions about personal attitudes and even gives up perceptions about government services. Moreover, the effective recovery rate of its Internet panel is not high, and after processing the data the sample is even less representative, so we select the data of CGSS 2015. Additionally, Li [61] concluded that the Internet would have an adverse effect on SFP through the latest data from CSS 2017 as well, which to a certain extent alleviates the timeliness issue in our research. Secondly, the measurement of variables is simple. Limited by the availability of data, for IU, we only ponder the frequency of Internet use and whether or not to use the Internet, while neglecting the type and purpose of Internet use (learning, socializing, entertainment, shopping, etc.) and the attitude towards the Internet and other aspects. Therefore, future research can continue to explore the differences and mechanisms in the impact of different abilities and purposes of using the Internet on SFP. Finally, the dependent variable SFP and the four metrics of the government trust are subjective, which may be affected by the situation and people’s mood at the time of the interview. Moreover, for some respondents, these problems are sensitive issues, and they may under-report their negative views for social desirability reasons, thus leading to measurement errors in the subjective variables. Then, subsequent studies should use longitudinal data over multiple years to mitigate measurement errors to some extent, while also exploring more specific and dynamic information.

## 6. Conclusions

The results of this study show that IU significantly reduce SFP among middle-aged Chinese and the impact is the complexion of intuitionism and rationalism. On the one hand, limited by their digital capabilities, middle-aged people are easily guided by online media in the process of using the Internet and form social attention and social emotions, which directly affects their evaluation of social fairness. On the other hand, in a society with redundant information, online media often pursue strange negative news. Timely, the government’s “gatekeeper” effect has taken a hit. All of these facts lead Chinese middle-aged people to lower their government trust when they go online. Low levels of trust in government inspires their lower SFP. It is worth mentioning that IU doesn’t increase the upward comparison of middle-aged Chinese, but instead expands the scope of their downward comparison, reduces their sense of relative deprivation, and indirectly improves their SFP to a certain extent.

## Figures and Tables

**Figure 1 ijerph-19-09855-f001:**
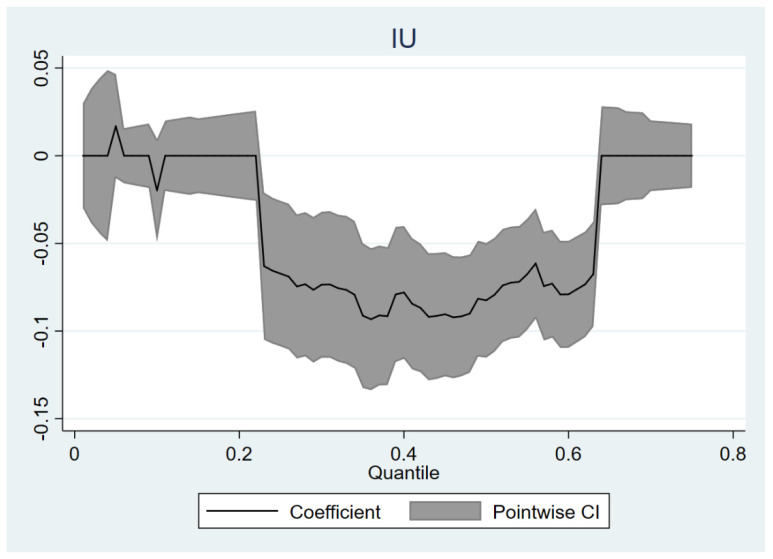
Quantile regression coefficient plot of IU.

**Figure 2 ijerph-19-09855-f002:**
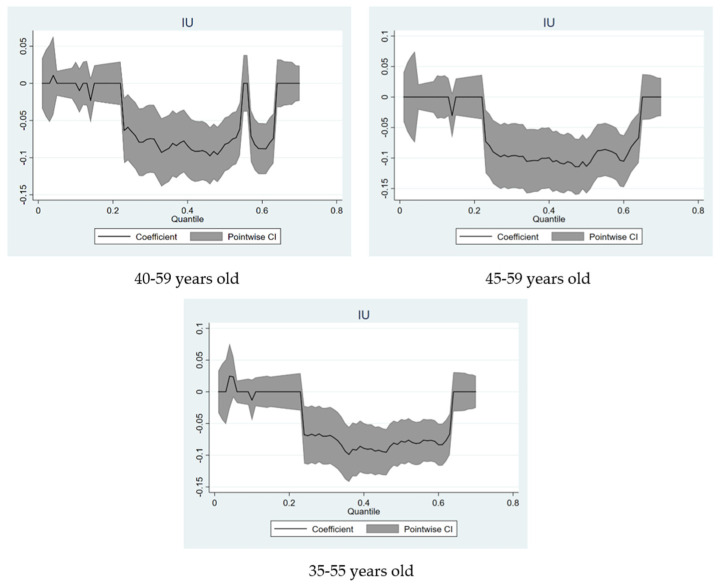
Quantile regression coefficients of IU under different age samples.

**Table 1 ijerph-19-09855-t001:** Descriptive statistics.

Variables	Mean	SD	Min	Max	N
SFP	3.109	1.020	1	5	4238
IU	2.364	1.561	1	5	4238
Age	47.345	6.828	35	59	4238
Gender	0.489	0.500	0	1	4238
Residence	0.571	0.495	0	1	4238
Region					
Western	0.220				934
Central	0.370				1569
Eastern	0.409				1735
Married	0.907	0.291	0	1	4238
Political affiliation	0.088	0.283	0	1	4238
Education	0.135	0.341	0	1	4238
Income	35,632.97	103,022.95	0	5,000,000	4238
Employed	0.510	0.500	0	1	4238
SWB	3.810	0.827	1	5	4238
Vertical comparison	0.113	0.316	0	1	4238
Expected comparison	0.358	0.480	0	1	4238
FSS	2.350	0.716	1	4	4238
PSS	0.345	0.475	0	1	4238
AFIU	2.354	0.965	1	5	4238
Ability	3.417	0.643	1	5	4238
Benevolence	3.169	0.939	0.2	5	1411
Consistency	2.985	0.885	0.5	5	1411
Impartiality	3.011	0.863	1	5	4238

**Table 2 ijerph-19-09855-t002:** OLS estimates of the effect of IU on SFP among the middle-aged.

Variables	Model 1	Model 2	Model 3
IU	−0.043 ***	−0.054 ***	−0.060 ***
	(−4.35)	(−4.31)	(−4.84)
Age		0.002	0.002
		(0.84)	(0.73)
Gender		0.025	0.030
		(0.82)	(1.00)
Residence		−0.148 ***	−0.141 ***
		(−4.12)	(−3.94)
Central regions (0 = western)		−0.077 *	−0.082 **
		(−1.92)	(−2.04)
Eastern regions (0= western)		−0.074 *	−0.069 *
		(−1.81)	(−1.69)
Married		−0.117 **	−0.139 ***
		(−2.29)	(−2.69)
Political affiliation		0.032	0.023
		(0.56)	(0.40)
Employed		−0.038	−0.040
		(−0.99)	(−1.05)
Education		0.129 ***	0.111 **
		(2.64)	(2.26)
Log of income		0.019 ***	0.016 **
		(2.72)	(2.20)
SWB		0.319 ***	0.299 ***
		(15.91)	(14.45)
Vertical comparison		−0.134 ***	−0.108 **
		(−2.58)	(−2.08)
Expected comparison		−0.348 ***	−0.308 ***
		(−10.73)	(−9.19)
PSS			−0.109 ***
			(−2.86)
FSS			−0.049 *
			(−1.89)
Constant	3.212 ***	2.120 ***	2.416 ***
	(111.21)	(13.21)	(13.42)
N	4238	4238	4238
R^2^	0.004	0.134	0.138

Note: In parentheses are the robust standard errors. * *p* < 0.1; ** *p* < 0.05; *** *p* < 0.01.

**Table 3 ijerph-19-09855-t003:** Tests for the validity of the instrument and the 2SLS estimates.

	SFP OLS	SFP OLS	IU 2SLS-First Stage	IU 2SLS-First Stage	SFP 2SLS-Second Stage	SFP 2SLS-Second Stage
IU	−0.047 ***	−0.054 ***			−0.091 ***	−0.096 ***
	(−3.53)	(−4.03)			(−2.85)	(−3.00)
AFIU	−0.033	−0.031	0.749 ***	0.741 ***		
	(−1.27)	(−1.22)	(27.34)	(27.23)		
PSS		−0.109 ***		−0.136 ***		−0.114 ***
		(−2.96)		(−3.22)		(−2.99)
FSS		−0.049 *		−0.160 ***		−0.056 **
		(−1.94)		(−5.57)		(−2.09)
Controls	Yes	Yes	Yes	Yes	Yes	Yes
Constant	2.142 ***	2.435 ***	2.400 ***	3.094 ***	2.246 ***	2.566 ***
	(13.58)	(13.61)	(13.43)	(15.45)	(11.90)	(11.72)
N	4238	4238	4238	4238	4238	4238
R^2^	0.134	0.138	0.505	0.512	0.132	0.136

Note: In parentheses are the robust standard errors. * *p* < 0.1; ** *p* < 0.05; *** *p* < 0.01.

**Table 4 ijerph-19-09855-t004:** Sensitivity analysis.

	OLS	IV-OProbit	IV-Probit	40–59	45–59	35–55
IU	−0.060 ***	−0.119 ***	−0.500 ***	−0.103 ***	−0.090 **	−0.079 **
	(−4.84)	(−3.27)	(−2.68)	(−2.95)	(−2.25)	(−2.26)
Controls	Yes	Yes	Yes	Yes	Yes	Yes
Constant	2.416 ***		−0.542	2.598 ***	2.641 ***	2.482 ***
	(13.42)		(−1.52)	(10.73)	(8.51)	(9.80)
N	4238	4238	3276	3565	2738	3603
R^2^	0.138			0.137	0.132	0.140

Note: in parentheses are the robust standard errors. ** *p* < 0.05; *** *p* < 0.01.

**Table 5 ijerph-19-09855-t005:** Analysis of the role path of social comparison.

	Family Subjective Status	Personal Subjective Status
Path C: estimates of the impact of IU on SFP		
	−0.054 ***	−0.054 ***
Controls	Yes	Yes
Path A: estimates of the impact of IU on social comparisons		
	−0.061 ***	−0.034 ***
Controls	Yes	Yes
Path B: estimates of the impact of social comparisons on SFP		
	−0.077 ***	−0.136 ***
Controls	Yes	Yes
Path C` (direct effect): the impact of IU on SFP, conditioning on the social comparisons		
	−0.058 ***	−0.058 ***
Controls	Yes	Yes
_bs_1	[0.0018713, 0.007921]	[0.0020974, 0.0077372]
_bs_2	[−0.0829642, −0.0341949]	[−0.0809748, −0.0341683]
N	4238	4238

Note: *** *p* < 0.01.

**Table 6 ijerph-19-09855-t006:** Heterogeneous effects (2SLS).

	Low Income	Middle Income	High Income
IU	−0.111 ***	0.021	−0.732 **
	(−2.77)	(0.36)	(−2.22)
Controls	Yes	Yes	Yes
Constant	2.673 ***	2.424 ***	7.759 ***
	(10.49)	(5.26)	(3.69)
N	3027	1073	138
R^2^	0.147	0.129	

Note: In parentheses are the robust standard errors. ** *p* < 0.05; *** *p* < 0.01.

**Table 7 ijerph-19-09855-t007:** Mechanism analysis.

	Ability	Benevolence	Consistency	Impartiality
Path C: estimates of the impact of IU on SFP				
	−0.060 ***	−0.058 ***	−0.058 ***	−0.060 ***
Controls	Yes	Yes	Yes	Yes
Path A: estimates of the impact of IU on intermediate factors				
	−0.052 ***	0.021	−0.026	−0.031 ***
Controls	Yes	Yes	Yes	Yes
Path B: estimates of the impact of intermediate factors on SFP				
	0.305 ***	0.183 ***	0.169 ***	0.178 ***
Controls	Yes	Yes	Yes	Yes
Path C` (direct effect): the impact of IU on SFP, conditioning on the intermediate factors				
	−0.045 ***	−0.062 ***	−0.054 ***	−0.055 ***
Controls	Yes	Yes	Yes	Yes
_bs_1	[−0.0211925, −0.0103364]	[−0.0044072, 0.0118547]	[−0.0113208, 0.0026979]	[−0.0095478, −0.0014611]
_bs_2	[−0.0663405, −0.0196704]	[−0.1009427, −0.023814]	[−0.0981255, −0.0116527]	[−0.0787886, −0.0301208]
N	4238	1411	1411	4238

Note: *** *p* < 0.01.

## Data Availability

This paper uses data from the China General Social Survey in 2015. CGSS started from 2003 and is the earliest national, comprehensive and continuous academic survey project in China, which systematically collects data at multiple levels of society, community, household and individual.

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
