# Peer review of "Rationalism or Intuitionism: How Does Internet Use Affect the Perceptions of Social Fairness among Middle-Aged Groups in China?"

_ijerph, 2022, doi:10.3390/ijerph19169855_

Round 1
Reviewer 1 Report
Overall, the text is interesting and valuable. However, the article may still be improved. In particular, it is worth clarifying whether the conclusions concern only Chinese society or are universal. It is also worth clearly indicating the most important directions of future research, which should be a continuation of the presented results.
The article is devoted to research on the impact of IU (internet use) on SFP (perception of social fairness) and identifies interesting relationships (which are nconsistent with some previous studies). To some extent, the problem belongs to the area of ​​digital health, so it can be published in ijerph journal (however, the issues raised relate more to psychology and sociology).
The topic is important due to the detection of dependencies contradictory to the results of the previously conducted research. However, the content of the article leaves a certain dissatisfaction. Not everything is clearly explained. The basic problem is whether we are dealing with a local effect or a universal dependency. If the phenomenon is local, it is necessary to explain under what specific conditions it may occur. What determines the current specificity of Chinese society?
The publications of other authors that were cited in the literature review do not necessarily describe phenomena adequate to the conditions of Chinese society (if we assume that Chinese society is unique). This issue should be clarified because the research model uses the insights of other authors describing phenomena from a different cultural circle. Additionally, certain statements are made without citing the relevant literature sources.
As for the methodology, apart from the issue of distinguishing between local and universal observations, doubts concern the questions of the survey itself. The answer options chosen by the respondents are partly an expression of their subjective feelings and not of facts. Certain doubts also concern whether the answers to the questions allow for the construction of sufficiently precise indicators describing the studied phenomena. Perhaps the authors could provide additional clarifications regarding these doubts.
As part of the proposed model, the conclusions correspond to the conducted research procedure, however, some doubts described above remain.
The references are appropriate, but it is worth increasing the precision in referring to the works of other authors by indicating what is local and what is universal.
Author Response
Dear reviewer:
Thank you for your letter and for the reviewers' comments concerning our manuscript entitled “Rationalism or Intuitionism:How does Internet use affect the perceptions of social fairness among middle-aged groups in China?” (ID: ijerph-1833259). Those comments are all valuable and very helpful for revising and improving our paper, as well as the important guiding significance to our researches. We have studied comments carefully and have made correction which we hope meet with approval. Revised portion are marked in the paper using the “Track Changes” function. The main correction in the paper and the responds to the reviewers' comments are as following: Responds to the reviewers' comments:
Reviewer #1:
- The comment: The basic problem is whether we are dealing with a local effect or a universal dependency. If the phenomenon is local, it is necessary to explain under what specific conditions it may occur. What determines the current specificity of Chinese society?
Our response: In response to the reviewer's comments, we hereby clarify that this paper deal with a Chinese local effect. The data we used is the Chinese General Social Survey 2015, and the database we employed only contains Chinese residents, so the results of this paper cannot be generalized to other regions outside of China.
We believe that there are two specificities in China. First, the digital ability of netizens is bad. Second, the Chinese government has a huge influence on people's lives. According to the 37th China Internet Development Statistical Report, as of December 2015, China's digital Internet penetration rate was 50.3%, a low level. According to the sample data for this paper, in 2015, nearly half of people aged 35-59 never used the Internet, and about 10% used the Internet occasionally. In addition, only 13.5% of middle-aged people have a college education level or above. These data show that Chinese middle-aged people have poor digital skills, and are easily guided by the media that pursues curious information which in turn condense social attention and generate negative emotions. On the other hand, the government occupies a very important position in the lives of Chinese people, and people's government trust greatly inspires their evaluation of society. With the development of internet, it has impacted the government's "gatekeeper" effect, and the government's ability to intercept negative information has declined. But the government's thinking has not changed, and it is still secretive about negative information. As a result, people have less and less trust in government and social fairness perception. At present, China's Internet penetration rate has been greatly improved (the 49th China Internet Development Statistical Report shows that the Internet penetration rate has risen to 73.0% by the end of 2021), and the first threshold for entering the Internet is gradually being eliminated, but the second threshold for using the Internet is still severe, and the negative guiding effect of the Internet still exists. At the same time, with the rise of Internet penetration, it is more difficult to maintain the "gatekeeper" of the government, and people's trust in the government and evaluation of the society have deteriorated.
2. The comment: The research model uses the insights of other authors describing phenomena from a different cultural circle which do not necessarily describe phenomena adequate to the conditions of Chinese society (if we assume that Chinese society is unique). Additionally, certain statements are made without citing the relevant literature sources.
Our response: According to the reviewer's comments, we have replaced inappropriate references which come from different cultural circles, or have explained the applicability of other cultural references in China. At the same time, we also supplemented the relevant literature citations to the certain statements. For example, we have added the references to "structural theory" and "relative deprivation theory".
3. The comment: As for the methodology, the answer options chosen by the respondents are partly an expression of their subjective feelings and not of facts. Certain doubts also concern whether the answers to the questions allow for the construction of sufficiently precise.
Our response: It is really true as reviewer suggested that the respondents may answer to the subjective questions uncertainly for the interference of the interview situation and mood and they may under-report their negative views for social desirability reasons which will cause measurement errors. In this regard, we supplement this in the shortcomings of the paper, and look forward to using balanced panel data to mitigate this deficiency in future studies.
4. The comment: It is worth clearly indicating the most important directions of future research, which should be a continuation of the presented results.
Our response: We illustrate our expectations for future research in view of the shortcomings of this paper, including the following two points: first, to explore the causal effects of different types and abilities of Internet usage on the social fairness perception of different groups (such as women and men); second, to use cross-year tracking data to explore the trend of the Internet's impact on people's social fairness perception over time which can also alleviate the measurement error of subjective variables.
Reviewer 2 Report
I recommend to use the extensive form of the phrase and in the parenthesis the acronym. For example, Project Management (PM) first time in the text, and next you can easily use only PM.
Also, please present the definitions of the SFP, IU, OLS, CGSS (if is the case) at the beginning of the paper.
You translated `SFP` as ` perception of social fairness` in this case I suppose the acronym is `PSF`. Please clarify.
Kindly, organize the presentation of the results as clearly as possible. There are long sentences and at some point you lose the meaning of what you wanted to transmit to the reader. There are also missing some punctuation marks.
There are quotes that are not commented upon them, please review.

Author Response
Dear reviewer:
Thank you for your letter and for the reviewers' comments concerning our manuscript entitled “Rationalism or Intuitionism:How does Internet use affect the perceptions of social fairness among middle-aged groups in China?” (ID: ijerph-1833259). Those comments are all valuable and very helpful for revising and improving our paper, as well as the important guiding significance to our researches. We have studied comments carefully and have made correction which we hope meet with approval. Revised portion are marked in the paper using the “Track Changes” function. The main correction in the paper and the responds to the reviewers' comments are as following:
Responds to the reviewers' comments:
Reviewer #2:
- The comment: I recommend to use the extensive form of the phrase and in the parenthesis the acronym.
Our response: We're sorry that we overlooked the use of abbreviations for some complex words. In this revised paper, we abbreviate instrumental variable as "IV", "Family subjective status" as "FSS", "Personal subjective status" as "PSS", and "average frequency of Internet use in the same community" as "AFIU".
- The comment: Please present the definitions of the SFP, IU, OLS, CGSS (if is the case) at the beginning of the paper.
Our response: We are sorry that the full form of these words was not given at the beginning of the paper. We have corrected this error and clarify that the SFP means “social fairness perception”, the IU means “internet use”, the OLS means “Ordinary Least Square”, and the CGSS means “Chinese General Social Survey”.
- The comment: Please clarify thetranslation of “SFP”.
Our response: We are very sorry for the incorrect use of the acronym SFP. Here we clarify that the dependent variable of the paper is “social fairness perception”, and its acronym is “SFP”.
- The comment: Organize the presentation of the results as clearly as possible. There are long sentences and at some point, you lose the meaning of what you wanted to transmit to the reader. There are also missing some punctuation marks.
Our response: According to the reviewer's suggestion, we have reorganized the language in this part to make it as concise as possible. At the same time, we also added some missing punctuation.
- The comment: There are quotes that are not commented upon them, please review.
Our response: We are sorry to have made such a mistake. In the revised paper, we have matched all citations to references to ensure that nothing is left out.
Special thanks to you for your good comments.

Round 2
Reviewer 2 Report
Dear Authors, Congratulations on the revised version of the paper.
Kind regards,